# Providing Comprehensive Dietary Fatty Acid Profiling from Saturates to Polyunsaturates with the Malaysia Lipid Study-Food Frequency Questionnaire: Validation Using the Triads Approach

**DOI:** 10.3390/nu13010120

**Published:** 2020-12-31

**Authors:** Zu-Wei Yeak, Khun-Aik Chuah, Choon-Heen Tan, Menagah Ezhumalai, Karuthan Chinna, Kalyana Sundram, Tilakavati Karupaiah

**Affiliations:** 1Nutrition Program, Faculty of Health Sciences, Universiti Kebangsaan Malaysia, Kuala Lumpur 50300, Malaysia; yeak_wei@hotmail.com (Z.-W.Y.); cruise_chuah@hotmail.com (K.-A.C.); choonheen@gmail.com (C.-H.T.); 2Department of Biomedical Sciences, Faculty of Medicine and Health Sciences, University Putra Malaysia, Serdang 43400, Malaysia; emenagah@yahoo.com; 3School of Medicine, Faculty of Health and Medical Sciences, Taylor’s University, Subang Jaya 47500, Malaysia; karuthan@gmail.com; 4Malaysian Palm Oil Council, Menara Axis, Petaling Jaya 46100, Malaysia; kalyana@mpoc.org.my; 5School of Biosciences, Faculty of Health and Medical Sciences, Taylor’s University, Subang Jaya 47500, Malaysia

**Keywords:** food frequency questionnaire, dietary records, biomarkers, dietary assessment, fatty acid composition, comparative validity

## Abstract

To address limited food frequency questionnaire (FFQ) capacity in public health monitoring in Malaysia, we aimed to develop a semi-quantitative FFQ for an adult multiethnic population for comprehensive fatty acid (FA) profiling inclusive of saturated (SFA), monounsaturated (MUFA), polyunsaturated fatty acids (PUFA), PUFA:SFA ratio, trans fatty acids, omega-3 and omega-6 FAs. A 240-food itemed FFQ used diet records (DR) of Malaysia Lipid Study (MLS) participants and fatty acid composition database from laboratory analyzed foods. The developed MLS-FFQ underwent face and content validation before relative validation in a free-living population (*n* = 114). Validation was facilitated for macronutrient data comparisons between DR and FFQ via Spearman’s correlation coefficient analyses; and for fatty acid composition data by independent pairing of DR, FFQ and plasma triglyceride using the triads method. Moderate correlation between dietary methods was obtained for macronutrients and FAs (*r* = 0.225–0.457, *p* < 0.05) except for ω-3 FAs, presenting good agreement with grossly misclassified nutrients <10%. For fatty acid composition data, the magnitude of validity coefficients (*z*) for SFA, PUFA, PUFA:SFA ratios and ω-6 FAs by all 3 methods were not significantly different (*p* > 0.05). In conclusion, the MLS-FFQ was shown to be a valid tool to assess population dietary intakes.

## 1. Introduction

Containment of dietary risks in population diets is been centered in global health initiatives for preventing obesity and the non-communicable diseases (NCDs) [1]. The Global Burden of Disease assessment indicates cardiovascular diseases recorded the highest age-standardised proportions of disability-adjusted life years attributable to dietary risks and diet-related deaths [2]. For better global health gains, the World Health Organization has now advised a population prevention strategy for countries where dietary guidelines as well as nutrition labelling should target limitation of saturated fatty acids (SFAs) and elimination of trans fatty acids (TFAs) in foods [3]. Within countries, the impetus for a successful population health nutrition strategy depends on timely dietary risks assessment for NCDs, both at national and community levels.

The use of the FFQ to assess population nutrition is the instrument of choice as it is cost effective, enables larger sampling and provides greater geographical coverage compared to limited daily diet recalls [4,5,6]. Moreover, regular FFQ administration within a population enables tracking temporal changes in food consumption. In this context, the nature of food consumption in the global food system is rapidly changing from traditional foods to ultra-processed and fast foods [7] which particularly affect the quality of fat consumed. This defines not just SFAs, polyunsaturated (PUFA) and monounsaturated (MUFA) fatty acids (FA) classes but importantly link to *trans*-fatty acids (TFA) and foods that are significant contributors to TFA intake [8].

The developing trend in FFQs that assess population FA intakes, selectively focus on PUFA assessment given the benefits for human health from omega-6 PUFA in reducing LDL cholesterol [9] and the omega-3 PUFA subclasses in reducing blood triglycerides and inflammation [10,11], but higher SFA and TFA intakes in populations have been associated with higher coronary heart disease (CHD) risk due to atherosclerosis [12,13]. For effective population nutrition assessment, FFQs need to provide for comprehensive assessment of all three FA subclasses as they are critically distributed in specific food groups and together constitute the quality of fats consumed in population diets. Comprehensive FA assessment together with macronutrient and energy profiling in an FFQ is therefore called for, to inform dietary guidelines appropriate for population nutrition action on NCDs prevention.

In the context of NCDs prevention [1] and dietary risks associated with disability-adjusted life years [14], an FFQ providing comprehensive FA profiling needs to be validated against biological markers of FAs. In particular, the fatty acid composition of plasma lipids, erythrocytes and adipose tissues have been referenced for the validation of newly developed FFQs [4,5,6]. FA profiles of plasma lipids and adipose tissues reflects the type of current habitual dietary fat intake and this could be useful to estimate the type of fats proportionally consumed by an individual [15].

Comparatively in Malaysia, the FFQ used in the national health survey [16] lacks the required fatty acid composition database, whilst another FFQ [17] provided assessment of SFA, omega-6 PUFAs and MUFAs but ignored TFAs and omega-3 FAs which potentiate either health risks or benefits. Moreover, this FFQ was not validated against blood FA biomarkers and only applicable to one ethnic group; whereas Malaysia is home to 3 ethnic groups with multicultural food practices that bear considerably influence on the food diet matrix [18]. Recently, the Malaysia Lipid Study (MLS) indicated that the atherogenic and cardiometabolic burden of a disease-free urban Malaysian population consuming high carbohydrate-high fat diets was very high [19]. Therefore, in Malaysia, there is a critical need to develop an FFQ tool with a validated fatty acid composition database to enable comprehensive FA assessment of population nutrition characteristics related to dietary risks for NCDs.

Our group aimed to develop a purposive semiquantitative FFQ relevant to capturing food data with FAC composition for the Malaysian population. The development of the FFQ will draw from the food recall database of the MLS [19]. The food listing in the MLS-FFQ will aim not only to include foods that are rich in fat but allow for assessment of a population’s nutrients of interest such as SFA, TFAs, and ω-3 and ω-6 PUFAs along with energy and macronutrients. Further, the MLS-FFQ would be designed to recognize variation in FA classes within food groups such as fish and cooking methods which greatly differ from Western techniques. Asian food recipes mostly utilize ‘stove top cooking’ techniques that incorporate vegetable oils and fats at the start of the cooking cycle [18]. Lastly, in terms of biological validation the tool would be compared with plasma triglyceride FAs using the triads approach [4,20,21]. The method calls for triangulation of FA data from dietary records (DR) and FFQ assessments with plasma FAs of subjects yielding validity coefficients of independent pairings that may be ranked by strength of associations, as well as indicate agreement between methods if not significantly different.

## 2. Materials and Methods

### 2.1. Study Design

The purpose of this research was to develop an FFQ instrument that enables characterization of both macronutrients and FA intakes in the adult Malaysian population. This study was structured into Phase 1 focusing on developing the FFQ, whilst Phase 2 progressed to face and content validation before relative validation of the finalized MLS-FFQ in a free-living population. The study received approval from the Medical Ethics Committee of UKM medical center (UKM.1.21.3/244/NN-2015-045). Subjects participating in the face and content validation of the MLS-FFQ, as well as the relative validation process provided inform consent.

### 2.2. Phase 1—Development of the Food Frequency Questionnaire (FFQ)

The development of the semiquantitative FFQ was a data-based approach [22] using dietary survey data (*n* = 577) collected through the MLS which was conducted between November 2012 to November 2013 in the urbanized Klang Valley region [19]. MLS subjects were healthy men and women, aged between 20 to 65 years old, with a racial representation of Malays (*n* = 288), Chinese (*n* = 203) and Indians (*n* = 146). Three 24-h dietary records (DRs) were collected per subject representing two weekdays and one weekend as per the MLS protocol. From this dataset, DRs of 450 subjects with equal representation of the three main ethnic groups were randomly selected. The sampling size was based on the minimum of 384 subjects required [23] for a total resident population of 6 million in the Klang Valley [24].

#### Food Frequency Questionnaire Development

[i] Construction of Food List—The construction of a food item list allowed database capacity for providing information on energy, macronutrients and fatty acid composition. After data entry completion of food items extracted from the DRs, a food item list was compiled by merging similar foods within categories as described by Block et al. [24]. The merging process for items within a food category allowed grouping similar foods with a similar cooking method. For example, ‘chicken cooked with soy sauce’, ‘chicken cooked with ginger’ and ‘chicken cooked with onion’ were grouped together under the ‘stir-fried chicken’ category. Merging, narrowed the total food items listed to 308 food items which were then ranked according to percentage contribution of each food item to total dietary fat intake calculated by using the formula [24] below:$$\%\;\mathrm{frequency}\;\mathrm{intake}\;\mathrm{contributed}\;\mathrm{item}=\frac{\mathrm{total}\;\mathrm{intake}\;\mathrm{of}\;\mathrm{specific}\;\mathrm{item}\;\times \;100}{\mathrm{total}\;\mathrm{intake}\;\mathrm{of}\;\mathrm{all}\;\mathrm{food}}$$

Overall, the food items forming the top 95% of contributors to fat intake were retained [25] in the FFQ, bringing the finalized number to 240 items.

[ii] Building Nutrient Composition—Food composition databases from the Malaysia Food Composition Table [26], Singapore Food Composition [27], USDA National Nutrient Database [28] and the ASEAN Food Composition Database [29], as well as food labels, were referenced to estimate proximate content of energy, carbohydrate, protein and fat for food items, in terms of absolute (g) weight as well as percent calories. However, these databases either lacked or had limited information on individual FAs as well as FA classes of locally consumed foods identified in the food list [26]. In particular, none of the referenced databases informed on TFA content. Laboratory analyses for fatty acid composition of the shortlisted food items [as described below] was, therefore, necessarily performed for this study by the researchers. Our laboratory had previous experience of performing these analyses [30,31,32].

In relevance to the nature of foods, samples were sourced from multiple food retail outlets such as street hawkers, food courts, restaurants and fast food outlets. The food retail sources were all located within the Klang Valley, representing an urban food environment. Each food item was purchased in duplicate from 3 different locations in the Klang Valley.

[iii] Food Fatty Acid Composition content—Collected food samples were homogenized together as described in the standard protocol for the *Malaysian Food Composition Database* [33]. For example, referring to the food item *roti canai*, samples were bought in duplicate from three different locations in the Klang Valley, all samples homogenized together before sampling two aliqouts for analyses. Crude fat in 10g of dried homogenous food samples were isolated by Soxhlet extraction [34], methylated to produce fatty acid methyl esters (FAME) by the Hitchcock and Hammond method [35], before subjecting to gas chromatography (GC) [35] to determine sample fatty acid composition profiles. The final GC values as per fatty acid composition profile for *roti canai* was taken as the mean of independent analysis of the aliquots for the relevant FA.

[iv] *Consumption patterns*—Additional information on the portion size and frequency of consumption of these food items on a daily, weekly, monthly or never/rarely used basis, were incorporated into the FFQ, with reporting based on common Malaysian household measures such as small bowl, medium bowl, cup, matchbox, glass, tablespoon and teaspoon [26,36]. The food listing (Appendix A) was categorized into 18 different food groups.

### 2.3. Phase 2—Validation of the FFQ

#### 2.3.1. Face and Content Validation

The FFQ underwent face and content validation with 20 volunteers, categorized in two groups, namely nutrition experts (*n* = 10) and lay persons (*n* = 10). Focus group discussions were conducted separately for these two groups. An information sheet was presented to participants explaining the format and process of the FFQ, and participant queries were facilitated by the researchers. Issues regarding the instructions were documented. Discussion for each group took approximately an hour. After resolving queries, participants were invited to self-administer the FFQ. The duration to complete the FFQ was timed with participants taking between 45 to 60 min to complete the FFQ. Upon completion of the FFQ, participants were invited to provide their feedback via an evaluation form as per (i) identity/familiarity of the food items, (ii) portion size, (iii) relevance to dietary practice, (iv) clarity of questionnaire (layout, colour, font and size) and (v) flow of the FFQ with these items rated on a 5-point Likert scale (1 = very poor, 2 = poor, 3 = fair, 4 = good and 5 = very good). Open-ended comments were invited if the score was below ‘fair’. Based on this evaluation, the FFQ was further revised. The finalized FFQ, now termed as the MLS-FFQ is available as a supplement (Appendix A).

#### 2.3.2. Relative Validation in a Free-Living Population

Community health screenings were initially conducted for 543 participants in the validation phase. Of these, 180 subjects became eligible to undergo a medical examination fulfilling criteria for age (20 to 60yrs old), free living, free of chronic disease and medication treatment, excessive alcohol consumption (>2 servings daily) and not on special dietary regimens or supplementation. Of 124 who participated in the medical examination, 114 subjects qualified to participate in the MLS-FFQ validation study. The study flow is represented in Figure 1.

Participants were provided DRs for two weekdays and one weekend to serve as a reference method to assess the validity of the FFQ [17,37]. After completion of DRs, subjects were probed for missing information in their records. After this interview, the MLS-FFQ was provided to respondents to self-administer, while those with limited literacy completed the MLS-FFQ through a face-to-face interview lasting ~45 min. Completed FFQs were collected and incomplete information immediately verified with respondents by the researcher.

#### 2.3.3. Sampling Size

The sampling size for the validation protocol was based on the formula of Hulley et al. [38]. With a correlation coefficient for total fat intake assumed to be 0.27 based on an earlier study [17] and a two-tailed significant level set at 0.05, a sample size of 103 subjects was determined as adequate to provide 80% power.

### 2.4. Fatty Acid Composition Analysis

[i] *Blood*
*fatty acid composition content*—At the time of the dietary interviews, a-5mL fasted blood sample was collected from each subject into a Vacutainer^®^ tube (Becton Dickinson Vacutainer, NJ, USA) containing EDTA (0.117mL of 15% EDTA). Samples were immediately centrifuged at 3000 *g* for 10 min (Sigma 3K12 B. Sigma Laborzentrifugen GmbH, Osterode am Harz, Germany) and obtained plasma stored at −80 °C until required. Lipids were extracted by Folch’s method [39] with 2 mL plasma and dried. Dried lipids were reconstituted with chloroform and then subjected to thin layer chromatography to isolate triglycerides (TGs). These TGs were methylated to produce FAME [35].

[ii] Gas chromatography—FAME from plasma and food samples were dissolved into small volumes of hexane prior to analysis by GC using the Shimadzu GC-2010 (Shimadzu Corporation, Tokyo, Japan) equipped with a flame ionization detector, a Shimadzu AOC-20i auto injector (Shimadzu) and a SP™—2560 capillary column (0.25 mm id × 100 m, 20 µm film thickness) from Supelco Inc (Bellefonte, PA, USA). Helium was used as the carrier gas.

Chromatographic spectra produced from FAME were compared with relative retention times of FAME of commercial standards carrying 99% purity (Supelco™). The *cis/trans* identifications were confirmed with the Supelco 37-component FAME Mix (47885-U) ranging from C_4_ to C_24_ run as an external standard. Additionally, linoleic acid methyl ester (C18:2) isomer mix (Cat.No.47791) and linolenic acid methyl ester (C18:3) isomer mix (Cat.No.47792) standards were also run under the same conditions. GC peak areas for methyl esters of identified FAs were reported as percent (%) fatty acid composition. Individual FAs in plasma TG and diets were expressed as percentage composition by weight.

### 2.5. Data Computation

[i] Fatty acid composition values of food items determined by laboratory analyses were embedded into a spreadsheet (Microsoft Office Excel) allowing for calculation of individual FA values of foods reported in the DRs by subjects.

[ii] For each subject, their DRs were analysed using Nutritionist Pro™ (Axxya Systems LLC, Stafford, TX, USA) to determine energy, carbohydrate, fat and protein intake. This data was exported to Microsoft Office Excel format where it was harmonized with individual fatty acid intake data from [i] to complete the dietary intake profile from DR.

[iii] A calculator using Microsoft Office Excel was developed for 18 food groupings in the MLS-FFQ based on harmonized energy, carbohydrate, fat, protein and fatty acid composition data. Conversion factors of 1.00, 0.14 and 0.03 were applied respectively for frequency of consumption of food items per day, per week and per month.

### 2.6. Statistical Analysis

The final data analysis included data from 108 subjects after exclusion for incomplete FFQ or DR (*n* = 2) and extreme dieting (*n* = 4). No misreporters were excluded in this study. This is to ensure the FFQ was able to detect misreporters in accordance with the DR. The underreporting phenomenon was expected as the prevalence of underreporting was estimated to be 64% for diet records and up to 88% for diet recall due to misreporting by subjects or misinterpretation by interviewers [40].

Mean and standard deviation, together with median and inter-quartile range (IQR), were calculated for nutrient intakes estimated from both the FFQ and DR. The validation of the FFQ was assessed using the methods of correlation coefficient and agreement. Spearman correlation coefficients were computed to assess the association between the two dietary tools. Wilcoxon signed-rank test was performed to compare the mean nutrient intakes estimated by FFQ and DR. Cross-quartile classification was used to detect if the FFQ and the reference method were able to discriminate subjects to the correct quartile of nutrient intake level respectively.

Using the Fisher *r*-to-*z* transformation of FA values determined by the triads ap-proach [4,20,21], validity coefficients (*ρ*) were computed for independent pairing of methods (FFQ, DR and TG). Based on this validity coefficient, the magnitude of agreement between each pairing was assessed as follows: ≥0.8 = high, 0.4–0.79 = moderate, 0.2–0.39 = low, <0.2 very low. This was followed by computing a *z* value to assess significance of the difference between three correlation coefficients, *ra, rb* and *rc*. The level of significance *p* > 0.05 for *z* was taken as agreement between methods. Additionally, the Bland and Altman plot [41] was also used for the purpose of testing agreement. All data was analyzed using SPSS 16.0 (IBM Corp, Armonk, New York, NY, USA). Statistical significance was set at *p <* 0.05.

## 3. Results

### 3.1. Subjects’ Characteristics

In total, 108 subjects comprising 59.3% females, completed the study. The mean age of the subjects was 35.4 ± 9.2 years old with an equitable ethnic distribution of 33.3% Malay, 32.4% Chinese and 34.3% Indian and overall mean BMI of 26.5 ± 5.6 kg/m^2^ (Table 1).

### 3.2. Validity Test between FFQ and Reference Methods

Generally, there was agreement (*p* > 0.05) of total fat intake, SFA, TFA and MUFA assessments between DR and FFQ methods (Table 2). Lack of agreement with mean differences exceeding 10%, was noted for total PUFA (% diff = 11.6), P/S ratio (% diff = 15.0) and ω-3 intakes (% diff = 26.0), with reported values been significantly higher by FFQ assessment compared to DR. Supplementary data is also provided for energy and macronutrients’ intake comparisons between DR and FFQ methods indicating good agreement and validity (Appendix A).

Between FFQ and DR comparisons (Table 3), all nutrients except ω-3 FA, showed significant correlations (*p* < 0.05). When comparing FAs in FFQ with TG as a biomarker, only PUFA (*r* = 0.192), P/S ratio (*r* = 0.0.245) and ω-6 FAs showed significant correlation (*p* < 0.05); whereas between DR and TG all FAs except for MUFA and TFA intake bore significant correlation (all *p* < 0.05).

Cross-quartile classification of calculated nutrients as per subject intakes’ (Table 4) indicated good agreement between FFQ and DR (<10% of gross misclassification) was possible for all nutrients. Further, with biomarker TG as a comparator for FAs good agreement with FFQ was present for percent SFA (4.6%), PUFA (8.3%), and ω-6 FAs (7.4%); whereas with DR good agreement was possible for only percent PUFA (5.6%), TFA (9.3%), ω-3 (6.5%) and ω-6 (8.3%) FAs.

Associations between pairwise differences between the triad formed from FFQ, DR and TG assessments were tested using Fisher *r* to *z* data transformation. As shown in Table 5, high-validity coefficients were observed between FFQ and TG (*b* = 0.98) for MUFA; and between DR and FFQ (*a* = 0.93) for TFA measurements. All three pairwise comparisons between methods yielded moderate-validity coefficients for percent SFA, PUFA, P/S ratios and ω-6 FAs. There was also moderate-validity (*ra* = 0.67) between DR and FFQ for percent ω-3 FAs. There were no significant differences in the magnitude of validity coefficients (*z*) for SFA, PUFA, P/S ratios and ω-6 FAs by all 3 methods. For MUFA, DR was significantly different from FFQ and TG. For ω-3 FAs, FFQ was significantly different from DR and TG. All 3 methods were significantly different from each other for TFA assessment.

No bias was observed for total energy, macronutrients’ intakes, SFA, TFA, and ω-6 FA between FFQ and DR as indicated by Bland-Altman plots (Appendix A). As regards FAC, bias was observed between MUFA, PUFA, P/S ratio and ω-3 FA.

## 4. Discussion

Frequent national and community-level nutritional surveys to monitor a population’s health risks related to food consumption, are required as a public health strategy for the prevention of NCDs. In context, an FFQ designed for comprehensive FA assessment coupled with macronutrient and energy profiling is critical to inform dietary guidelines appropriate for population nutrition action on NCDs prevention. The FFQ has been trialed in many countries to overcome the costs, time and logistics constraints in this manpower-intensive exercise [42]. Importantly, it is recognized [43] that an FFQ instrument has the capacity for continuous improvement as per refinement, modification and evaluation. Accordingly, the right database of locally consumed foods that enable assessment of SFAs, PUFAs, P/S, ω-6 and ω-3 FAs, and TFAs intakes coupled with energy and macronutrients’ profiling is crucial for an FFQ designed for population risk assessment. In Malaysia population surveys are conducted every decade with an FFQ [44] lacking the necessary criteria for providing comprehensive nutrient assessment for monitoring NCDs risk.

Our study demonstrated the newly designed semi-quantitative MLS-FFQ is suitable to assess habitual nutrient intakes of multiethnic groups living in urban Malaysia. More than 70% of Malaysians live in urban built environments [23], and the food database was fittingly drawn from an urban-living cohort, the MLS study subjects [19] with adequate multiethnic representation. The validation process was also undertaken with equal ethnic representation (33.3% Malay, 32.4% Chinese and 34.3% Indian). Adequate population representation is an important criterion in FFQ usability so as to enable detection of adequate and appropriate food intakes and patterns related to chronic disease risk or malnutrition [45]. Adequate ethnic representation was not seen for previous FFQs [17,44] used in Malaysia as they were developed for the Malay population.

The MLS-FFQ instrument included 240 foods which is acceptable for optimal food characterization of a population’s diet. Cade et al. [46] in reviewing over 200 FFQs noted food item inclusions ranged from a minimum five to 350 items. Comparatively in Malaysia, this 240-item MLS-FFQ included more food items then the 89-food item FFQ developed by Eng and Moy [17], who examined SFA, PUFA and MUFA intakes. An open-ended frequency format adopted for the MLS-FFQ, elicited more information items as opposed to the closed format [47]. Portion sizes cited in the MLS-FFQ were based on household measures and this is the recommendation to reduce errors in quantification of portion size by subjects [48]. Close-ended options of food preparation techniques of stir-frying, deep-frying, steaming, curry-based, soup or grilling were applied to this FFQ to increase instrument specificity as recommended [49]. Food preparation and cooking methods play a major role in forming derivatives of raw primary commodities [50] commonly consumed by a population, and such food composites were factored into the FFQ development. The food items in the MLS-FFQ are available in Malaysia throughout the whole year and been on the equator this region has access to continuous supply of foods. This implies that there is consistent food availability in Malaysia as regards any time period for the application of this FFQ.

Margretts and Nelson [51] encouraged prioritizing local food databases for analysing nutrient content in order to minimize error. For this reason, nutritional composition of foods was referenced from the Malaysian Food Composition Table [26] and Singapore Food Composition Table [27]. However, many local foods identified through the MLS food database lacked fatty acid composition information. As accuracy of data is reduced with food substitution, we undertook to conduct fatty acid composition analyses of 365 foods in our laboratory. Some of this fatty acid composition data has been published elsewhere [32,52]. Of particular interest, the MLS-FFQ was purposively designed to recognize variation in fatty acid classes within a food group. For example, for the fish category, fish species were grouped into low, moderate or high fat, fish of cold water or tropical sea origin, and also by cooking preparation method. These sub-categories within a food group differentiates PUFA class distribution as well as total fat content [46].

Validation of the FFQ is critical to check inaccuracies related to food identity, portion size and ingredients, as estimation of both macronutrients and FA profiles are increasingly important in determining associations between dietary factors and chronic diseases. In particular, the MLS-FFQ design features were as per recommendations raised in expert dietary guidelines which call for reductions in a population’s intake of SFA, limit TFAs and to increase PUFAs. In our study we first validated the MLS-FFQ against 3 days of diet records (DR). The most cited methodology to validate a dietary tool targeting fat intake assessment were by comparing FFQ against multiple days of diet recalls or diet records [45,53]. As opposed to this, biological markers with lipid plasma, erythrocytes or adipose tissues may serve as reference for the validation of FAs [4,5,6]. In this regard, we performed a second validation for FA intakes assessed by both FFQ and DR with plasma triglycerides (TG). We found that the MLS-FFQ adequately estimated true intake values for SFA, PUFA, P/S ratio and ω-6 FAs in agreement with TG and DR. As regards TFA, although the FFQ showed moderate correlation with TG and high correlation with DR, the overall true intake values estimated by all three methods was not similar. A plausible reason is a low consumption of TFA-rich foods in the Malaysian population due to higher palm oil consumption [19]. Similarly for ω-3 FAs true intake assessment was weak by all three methods, indicating circulating levels in TGs reflect short-term consumption of relevant foods [15,36]. Erythrocyte FAs are more likely to be a robust comparator for TFAs and ω-3 FAs [4,54]. However, findings indicated the MLS-FFQ was close to DR for assessing TFA and ω-3 FAs as the reference food database was shared. It is also observed that the MLS-FFQ showed moderate correlation to TG for MUFA assessment.

There are some limitations noted with this study. Firstly, there was exclusive reliance on the Soxhlet method in the extraction of crude fat from the food samples in this study. However, the Soxhlet extraction is commonly used and is a well recognized method for isolating lipids from a wide range of dried foods [55]. Secondly, the food items in the database of the MLS-FFQ are relevant to the current period of food consumption trends in this population, but food habits of a population change and new foods are introduced and become regularly consumed [56]. This is a limitation inherent to any FFQ [57]. This limitation may be overcome by updating the database of the FFQ with new foods. Margretts and Nelson [51] have opined that food databases should be updated continuously as new foods are developing and food preparation methods are changing over time. Further the MLS-FFQ is only suitable to be applied to this multiethnic population and limited to urban adults, as food choices are reflective of ethnicity, religious diversity and geographical conditions. However, the MLS-FFQ design is novel and would easily allow for adaptative use for South East Asian countries. We recently showed the adaptability of this MLS-FFQ in the development of the HD-FFQ suitable for an Asian dialysis population [58]. An important point is that the nutrient database used for the MLS-FFQ shared the same database as the reference method (DR) in order to minimise the variation and bias. Greenfield and Southgate [59] stated that different databases have different types of food included and variation in the methods of food analyses which introduces Type II error; an error where null hypothesis is false but fails to be rejected. An additional strength of the MLS-FFQ is the database was based on local foods.

## 5. Conclusions

In conclusion, the newly developed MLS-FFQ with a database of locally consumed foods allows assessment of SFAs, PUFAs, P/S ratio, and ω-6 FAs coupled with total energy, carbohydrates, fat and protein coupled with to be performed for population surveys in Malaysia. The application of this FFQ to population health surveys in Malaysia is suitable in terms of ethnic diversity, multicultural food choices and sociocultural practices.

## Figures and Tables

**Figure 1 nutrients-13-00120-f001:**
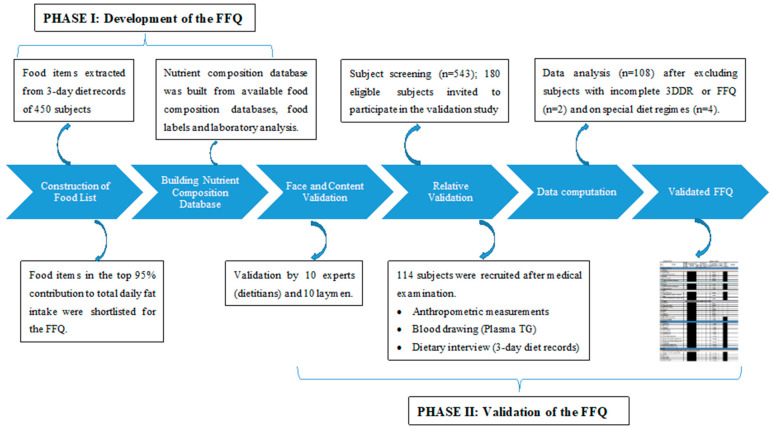
Process algorithm for development and validation of the MLS-FFQ.

**Table 1 nutrients-13-00120-t001:** Characteristics of subjects.

Characteristics (*n* = 108)	Values
Age, years	35.4 ± 9.2
Gender, male/female, *n*	44/64
Ethnicity, Malay/Chinese/ndian, *n*	36/35/37
Body Mass Index, kg/m^2^	26.5 ± 5.6
Mean energy intake by gender, kcal	
DR	
*Male*	2066 ± 410
*Female*	1499 ± 256
FFQ	
*Male*	2042 ± 553
*Female*	1616 ± 495
Mean energy intake by ethnicity, kcal	
DR	
*Malay*	1720 ± 365
*Chinese*	1908 ± 475
*Indian*	1571 ± 386
FFQ	
*Malay*	1812 ± 525
*Chinese*	1775 ± 597
*Indian*	1782 ± 566

Note: Data are presented as mean ± SD unless stated as *n*. Abbreviations: DR = 3-day dietary records; FFQ = food frequency questionnaires.

**Table 2 nutrients-13-00120-t002:** Total Fat and Fatty Acid Intake by Dietary Assessment Method.

Dietary Intake	DR		FFQ		Mean Difference (%)	*p*-Value *
	Mean ± SD	Median (IQR)	Mean ± SD	Median (IQR)		
Total Fat (g/day)	59.1 ± 20.4	54.9 (45.4, 71.7)	62.6 ± 23.4	58.1 (47.5, 74.7)	−5.9	0.230
SFA (g/day)	26.4 ± 9.1	24.5 (19.5, 31.9)	27.2 ± 10.5	25.4 (20.4, 32.5)	−3.0	0.690
MUFA (g/day)	23.3 ± 8.8	21.2 (17.0, 29.3)	25.3 ± 9.6	23.6 (19.4, 29.2)	−8.6	0.100
PUFA (g/day)	8.6 ± 3.3	8.1 (6.2, 11.1)	9.6 ± 3.8	8.7 (7.1, 11.6)	−11.6	**0.030**
TFA (g/day)	0.15 ± 0.10	0.14 (0.08, 0.20)	0.16 ± 0.08	0.15 (0.10, 0.19)	−6.7	0.420
P/S ratio	0.34 ± 0.11	0.33 (0.28, 0.38)	0.36 ± 0.11	0.34 (0.31, 0.38)	15.0	**0.003**
ω-3 (g/day)	0.50 ± 0.35	0.41 (0.26, 0.70)	0.63 ± 0.38	0.52 (0.39, 0.81)	−26.0	**0.002**
ω-6 (g/day)	8.1 ± 3.1	7.5 (5.8, 10.3)	9.0 ± 3.5	8.1 (6.6, 10.9)	−11.1	0.063

Note: * Comparison between DR and FFQ using Wilcoxon signed rank test, significant with *p*-value < 0.05 is bold format; Percentage mean difference was individually computed using the formula (FFQ-DR)/DR×100). Abbreviations: DR = 3-day dietary records; g/day = gram per day; FFQ = food frequency questionnaires; IQR = inter quartile range; MUFA = monounsaturated fatty acid; ω-6 FAs = omega-6 acids; PUFA = polyunsaturated fatty acid; P/S = polyunsaturated fatty acid/saturated fatty acid; SFA = saturated fatty acid; TFA = trans fatty acid; ω-3 FAs = omega-3 fatty acids.

**Table 3 nutrients-13-00120-t003:** Correlations between FFQ, DR and plasma TG as per total fat and fatty acid profiles.

Nutrients	*r **	*p*-Value
FFQ versus DR		
Total Fat (g/day)	**0.357**	<0.001
SFA (g/day)	**0.346**	<0.001
MUFA (g/day)	**0.346**	<0.001
PUFA (g/day)	**0.234**	0.015
TFA (g/day)	**0.391**	<0.001
P/S ratio	**0.225**	0.019
ω-3 FAs (g/day)	0.095	0.329
ω-6 FAs (g/day)	**0.245**	0.011
FFQ versus plasma TG		
SFA (% fat)	0.184	0.057
MUFA (% fat)	−0.175	0.069
PUFA (% fat)	**0.192**	0.047
TFA (% fat)	−0.004	0.964
P/S ratio	**0.245**	0.011
ω-3 FAs (% fat)	0.098	0.315
ω-6 FAs (% fat)	**0.295**	0.007
DR versus plasma TG		
SFA (% fat)	**0.197**	0.041
MUFA (% fat)	−0.043	0.657
PUFA (% fat)	**0.223**	0.02
TFA (% fat)	0.174	0.072
P/S ratio	**0.205**	0.034
ω-3 FAs (% fat)	**0.272**	0.004
ω-6 FAs (% fat)	**0.241**	0.012

Note: * Spearman’s rank correlation coefficient values ‘r’ in bold are significant with *p*-value < 0.05 Abbreviations: g/day = gram per day; DR = 3-day dietary records; FFQ = food frequency questionnaires; MUFA = monounsaturated fatty acid; ω-3 FAs = omega-3 fatty acids; ω-6 FAs = omega-6 fatty acids; PUFA = polyunsaturated fatty acid; P/S = polyunsaturated fatty ac-id/saturated fatty acid; SFA = saturated fatty acid; TFA = trans fatty acid; TG = plasma triglyceride fatty acid biomarker.

**Table 4 nutrients-13-00120-t004:** Cross-quartile classifications for total fat and fatty acid profile comparisons between FFQ, DR and plasma TG.

Nutrients	Same Quartile (%)	Adjacent Quartile (%)	Grossly Misclassified (%)
FFQ versus DR			
Total Fat (g/day)	33.3	40.7	**5.6**
SFA (g/day)	34.3	38.9	**5.6**
MUFA (g/day)	28.7	43.5	**3.7**
PUFA (g/day)	25.0	43.5	**5.6**
TFA (g/day)	38.0	34.3	**8.3**
P/S ratio	33.3	38.9	**9.3**
ω-3 FAs (g/day)	21.3	70.4	**8.3**
ω-6 FAs (g/day)	23.1	71.3	**5.6**
FFQ versus Plasma TG			
SFA (% fat)	24.1	39.8	4.6
MUFA (% fat)	15.7	40.7	16.7
PUFA (% fat)	25.0	39.8	**8.3**
TFA (% fat)	30.6	34.3	12.0
P/S ratio	30.6	38.9	11.1
ω-3 FAs (% fat)	28.7	59.3	12.0
ω-6 FAs (% fat)	32.4	60.2	7.4
DR versus Plasma TG			
SFA (% fat)	19.4	65.8	14.8
MUFA (% fat)	23.2	64.8	12.0
PUFA (% fat)	36.1	58.3	**5.6**
TFA (% fat)	23.1	67.6	**9.3**
P/S ratio	23.2	57.4	13.4
ω-3 FAs (% fat)	32.4	61.1	**6.5**
ω-6 FAs (% fat)	29.6	62.1	**8.3**

Note: good agreement with <10% of gross misclassification was indicated by figures in bold. Abbreviations: DR = 3-day dietary records; en = energy; g/day = gram per day; FFQ = food fre-quency questionnaires; MUFA = monounsaturated fatty acid; ω-3 FAs = omega-3 fatty acids; ω-6 FAs = omega-6 fatty acids; PUFA = polyunsaturated fatty acid; P/S = polyunsaturated fatty ac-id/saturated fatty acid; SFA = saturated fatty acid; TFA = trans fatty acid; TG = plasma triglyceride fatty acid biomarker.

**Table 5 nutrients-13-00120-t005:** Validity coefficients for method of triads.

Fatty Acids Group	*ra*	*rb*	*rc*	*z* diff *(p*-Value)		
	*ρ* (DR vs. FFQ)	*ρ* (FFQ-TG)	*ρ* (DR-TG)	*ra—rb*	*ra—rc*	*rb—rc*
SFA (% fat)	0.49	0.52	0.46	−0.03 (0.800)	0.03 (0.772)	0.06 (0.587)
MUFA (% fat)	0.18	0.98	0.12	−0.80 (<0.001)	0.06 (0.654)	0.80 (<0.001)
PUFA (% fat)	0.37	0.48	0.56	−0.11 (0.331)	−0.19 (0.066)	−0.08 (0.384)
TFA (% fat)	0.93	0.38	0.10	0.55 (<0.001)	0.83 (<0.001)	0.28 (0.035)
P/S ratio	0.37	0.53	0.51	−0.16 (0.136)	−0.14 (0.199)	0.02 (0.838)
ω-3 FAs (% fat)	0.67	0.12	0.29	0.55 (<0.001)	0.38 (<0.001)	−0.17 (0.207)
ω-6 FAs (% fat)	0.44	0.48	0.53	−0.04 (0.715)	−0.09 (0.393)	−0.05 (0.625)

Note: Fisher *r* to *z* transformation with *ra* = correlation coefficient for validity between DR and FFQ; *rb* = correlation coefficient for validity between FFQ and TG; *rc* = correlation coefficient for validity between DR and TG; categorization as per ≥ 0.8 = high, 0.4–0.79 = moderate, 0.2–0.39 = low, <0.2 very low; and *z* diff for *p* > 0.05 indicates agreement. Abbreviations: DR = 3-day dietary records; en = energy; g/day = gram per day; FFQ = food frequency questionnaires; MUFA = monounsaturated fatty acid; ω-3 FAs = omega-3 fatty acids; ω-6 FAs = omega-6 fatty acids; PUFA = polyunsaturated fatty acid; P/S = polyunsaturated fatty acid/saturated fatty acid; SFA = saturated fatty acid; TFA = trans fatty acid; TG = plasma triglyceride fatty acid biomarker.

## Data Availability

The data presented in this study are available on request from the corresponding author. The fatty acid compositional data for the MLS-FFQ are not publicly available but restricted to public health stakeholders and researchers.

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
