# Peer review of "Providing Comprehensive Dietary Fatty Acid Profiling from Saturates to Polyunsaturates with the Malaysia Lipid Study-Food Frequency Questionnaire: Validation Using the Triads Approach"

_nutrients, 2020, doi:10.3390/nu13010120_

Round 1

Reviewer 1 Report

In this interesting study the authors assess the validity of a newly developed food frequency questionnaire aimed to capture total fat intake as well saturated, monounsaturated, polyunsaturated and trans fatty acid intake in adults in Malaysia.

The objective of the study is clearly formulated, the design of the study and the methods and data analysis are appropriate. The results are relevant and the discussion is thorough.

However, there are a few issues the authors will need to address:

Introduction

Page 2, lines 45-49

"successful population health nutrition strategy depends on timely surveys using food frequency questionnaires (FFQ) both at national and community levels which factor in assessment of dietary risks contributive to NCDs development".

The use of the FFQ to assess population nutrition is the instrument of choice as it enables larger sampling and greater geographical coverage compared to limited daily diet recalls [4-6].

FFQ are cost-efficient tools to be applied systematically. However, successful population health strategies do not depend on the use of FFQs. Such strategies should be based on efficient and timely surveillance systems able to provide reliable information. The methods used can vary. However, due to cost-effectiveness and ease of use, FFQs are often the tools of choice, provided the selected FFQ is a valid instrument.

Page 2, lines 82-83

"Our group aimed to develop a purposive semiquantitative FFQ relevant to capturing food data with FAC composition".

It would be more appropriate that the FFQ should be relevant to capturing usual FAC intake in the reference population.

Methods

page 3, lines 119-120

"Food items in the top 95% contribution to total energy and macronutrients’ intakes were 118 shortlisted for the FFQ [24]. The finalized food listing initially yielded 308 food items from the DRs of 450 MLS subjects. This food list was incorporated into the FFQ instrument as well as utilized to develop a food album to aid recall of food portion size consumed".

However, the FFQ food list cosists of 240 items. How did you reduce the food list from 308 food items to 240 food items? Which criteria did you apply?

page 3, lines 128-130:

"In particular, none of the referenced databases informed on TFA content. Laboratory analyses for FAC of the shortlisted food items [as described below] was, therefore, necessarily performed for this study by the researchers. Our laboratory had previous experience of performing these analyses [29-31].

You report you used Malaysian and Singapore Food Composition tables jointly with the USDA National Nutrient Database, which includes fatty acid information, and decided to conduct food composition analyses on Malaysian foods.

How did you sample the foods to analyse? Which was the procedure you followed? How many food samples from each food item did you analyse? From which location? In what season?

Face and Content Validation

Page 4, line 143

"The FFQ underwent face and content validation with a 20-member focus group"

How did you collected and analyzed the information from the focus group discussion?
Did you involve 20 people at a time for a focus group discussion? How long was the discussion?
Why did you decide to run a focus group discussion with 20 people instead of two smaller groups?

In addition, a few more questions:

  • How did you use the FFQ, in the context of a face-to-face interview? on-line? self-completed on paper? How long did it take to complete on average?
  • Which was the reference period to consider when using the FFQ?
  • Did you consider underreporting in data analysis?
  • Are you considering to test the reproducibility of the FFQ, ie. apply the FFQ on two different occasions on the same sample within a previuosly specified time frame?

Reviewer 2 Report

LINE 57: "lipid-lowering" is a inexact description of PUFA effects. Please, be more accurate.

LINES 82-92: you should explain the aim of your study but not details on what was done or the results obtained.

LINES 134-135: Soxhlet extraction is not adequate for fatty acids determination in foods (you should have used Folch or Bligh&Dyer methods), so you should mention this in the limitations of your study.

LINE 252 and TABLE 3: you compare FA ingested with FA present in plams TG. So, you should write "with fatty acids from plasmaTG as a biomarker". Same in TABLE 3.
